# The Fitter I Am, the Larger I Feel—The Vicious Circle of Physical Exercise in Anorexia Nervosa

**DOI:** 10.3390/nu14214507

**Published:** 2022-10-26

**Authors:** Laura Di Lodovico, Mouna Hanachi, Philibert Duriez, Philip Gorwood

**Affiliations:** 1Clinique des Maladies Mentales et de l’Encéphale (CMME), Hôpital Sainte-Anne, GHU Paris Psychiatrie et Neurosciences, 75014 Paris, France; 2Institute of Psychiatry and Neuroscience of Paris (IPNP), INSERM U1266, Université Paris Cité, 75006 Paris, France; 3Nutrition Unit, Paul Brousse University Hospital, Assistance Publique–Hôpitaux de Paris, 94800 Villejuif, France; 4Micalis Institute, INRA, AgroParisTech, Université Paris-Saclay, 78350 Jouy-en-Josas, France

**Keywords:** body image distortion, physical exercise, central coherence, cognitive rigidity

## Abstract

Introduction: Body image distortion is a core symptom of anorexia nervosa (AN), embodying dissatisfaction and overvaluation of body appearance and weight. Body image distortion is an important factor in the maintenance of weight loss behaviours such as compulsive physical exercise. Conversely, physical exercise seems to have an aggravating effect on body image in patients with AN, but the evidence is still poor. The aim of this study was to examine the relationship between body image distortion and physical exercise in AN in order to understand whether physical exercise may play a specific role in body image distortion beyond psychopathological severity. Methods: Forty patients with AN and 21 healthy controls were tested for body image distortion and different proxies of physical exercise. Univariate correlations tested the relationship between body image distortion and physical exercise in AN and control groups. Then, to experimentally assess the effect of exercise on body image distortion, participants were invited to rate their body image before and after a standardised effort test. Results: In the AN group, a correlation was found between physical activity and body image distortion (*p* < 0.01), which was still significant after controlling for psychopathological severity. No correlation was found in healthy controls. After a standardised effort, patients with AN had higher increases in body image distortion than healthy controls (almost 4 kg/m^2^ vs. 0.8 kg/m^2^) (*p* < 0.01). Discussion: Physical exercise may contribute to the distortion of body image in anorexia nervosa and explain the paradoxical augmentation of unhealthy exercise despite ongoing weight loss.

## 1. Introduction

Anorexia Nervosa (AN) is a severe and potentially fatal psychiatric illness characterised by a persistent fear of weight gain associated with an undue influence of body weight or shape on self-evaluation, leading to a state of emaciation that is often denied or underestimated. Thinness is followed by progressively compulsive behaviours like restrictive dieting and excessive physical exercise [1].

Body image disturbance (BID), considered one of the core characteristics of AN [2], is a multidimensional construct embodying dissatisfaction with and over-evaluation of body appearance and, to a lesser extent, weight and shape preoccupations [3]. The exact nature of this symptom is still unknown [2], but a consistent body of research suggests an impaired capacity to manipulate and process visuospatial material on a neurocognitive basis [4]. Impaired visuospatial abilities are nowadays considered one of the cognitive hallmarks of AN [5]. Even though results are still heterogeneous [6], superior attention to details in AN, at the expense of central coherence [2,6,7], is a constant feature in AN. The rigidity of frontal functioning also seems to play a role in the maintenance of an aberrant body image perception [8], with the lack of cognitive flexibility explaining the incapacity to update one’s body image over time despite progressive emaciation.

As a consequence of BID, patients with AN engage in weight loss strategies, such as food restriction, purging behaviours, and compulsive exercise [3], giving life to a vicious spiral in which ongoing emaciation corresponds to an aggravation rather than a limitation of the psychological and behavioural symptoms of AN [9].

Current evidence suggests a bidirectional relationship between BID and weight loss behaviours [10]: body dissatisfaction predicts restrictive eating practices [11,12], and body image distortion is enhanced by food ingestion in restrictive eaters [13]. Likewise, physical exercise serves as a means to cope with body dissatisfaction and to control shape and weight [14] but, in turn, seems to predict lower body esteem [13,15].

Unhealthy practice of physical exercise is a recognised core symptom of AN [1,9,14,15]. Patients with AN tend to engage in different forms of physical exercise, regardless of their malnourished state and the risks associated [16]. This drive to physical exercise represents a crucial concern for physicians and researchers involved in AN because of its paradoxical nature (i.e., its persistence, or even increase, along with illness evolution and weight loss [15]), its aggravating effect on malnutrition and prognosis [14], and its role in the development and maintenance of AN [16]. Physical exercise practised by patients with AN has reinforcing effects [17,18] that are associated with a particular increase in positive emotions. Its role as a maladaptive strategy for mood regulation partly accounts for its persistence, even at the most severe stages of the disease. A second aspect, explaining its persistence despite its counterindication, is the compulsive nature of this activity. At the initial stages of the disease, physical exercise is a goal-directed behaviour, often practised in the pursuit of thinness and emotional well-being. Over the course of the illness, physical exercise evolves into a habit-driven behaviour, is emancipated from its initial goals, and becomes repetitive, rigid, and hard to cease [9]. In parallel, an aggravation of BID is often found along with the illness’ evolution [1,3,6]. This parallel and progressive aggravation of BID and of compulsive physical exercise with the aggravation of AN may raise the question of a relationship between these two symptoms.

The paradoxical intensification of AN’s behavioural symptoms along the course of the illness could be explained by the hypothesis that BID is aggravated, rather than relieved, by weight loss behaviours, giving life to a pathological spiral whose mediators are still unknown. If a link between dieting behaviours and the aggravation of BID has been already shown, and a relationship between compulsive exercise and BID has already been found in body dysmorphic disorder [19], the role of physical exercise in BID in AN patients is still underexplored. Our previous research found that a single bout of physical exercise was sufficient to increase BID with a moderate effect size in patients with AN [17], and that BID had a significant correlation with the reported amount of exercise practised in a week. Replications of this important finding are still needed and should overcome the limitations encountered by this first research, such as the choice of the BID assessment tool. Most research on BID in AN uses a bidimensional, figural-drawing scale based on a limited range of figures, among which participants have to choose the one resembling theirs. Hence, a “coarse” response scale consisting of a limited number of response choices is employed to represent a nearly continuous or “fine” variable. Moreover, the role of potential confounders impacting BID, such as body mass index (BMI), illness duration, or AN psychopathology Eating Disorder Inventory-2 (EDI-2), was not explored.

Determining the relationship between exercise and BID may lead to a better understanding of the factors involved in this core symptom of AN and provide a sound explication of the paradoxical increase in physical exercise along with the progression of the illness. Hence, the aims of the present study are to understand if there is a specific correlation between BID and physical exercise in patients with AN and to ascertain the causal direction of this relationship using an experimental approach, i.e., to understand if physical exercise is able to aggravate BID in AN patients.

## 2. Methods

### 2.1. Study Design and Ethics

The present study utilises data collected as part of a monocentric, interventional open trial performed in the Clinique des Maladies Mentales et de l’Encéphale within the GHU Paris Psychiatrie et Neurosciences. The protocol trial encompassed an observational and an experimental phase.

All study procedures were approved on 22 September 2015 by the Comité de Protection des Personnes (CPP) Île-de-France III Ethics Committee, reference code: 3321. All study procedures were approved on 8 September 2015 by the Agence Nationale de Sécurité du Médicament et des Produits de Santé (ANSM)*,* reference code: 15108B-31. EudraCT code: 2015A01194-45.

In accordance with the Helsinki Declaration, written informed consent was obtained from each participant before inclusion. The trial was registered at clinicaltrials.gov with the identifier: NCT02995226.

### 2.2. Population

A group of patients with AN and a group of healthy controls (HC) were recruited. Inclusion criteria for the AN group were: a diagnosis of AN according to the Diagnostic and Statistical Manual of Mental Disorders (DSM-5) criteria [20], female sex, age between 18 and 50 years, and BMI between 14 and 17.5 kg/m^2^.

Inclusion criteria for the HC group were: female sex, age between 18 and 50 years, BMI > 17.5 kg/m^2^, and no personal history of eating disorder as screened using the Mini International Neuropsychiatric Interview (MINI) 7.0.2 for DSM-5.

Since the protocol envisaged a standardised effort test (already described in [17]), exclusion criteria for all groups were any contraindication to the practice of an intensive sport, i.e., personal history of hypertension, stroke, phlebitis, and/or heart attack; a first-degree family history of stroke at the age of <45 years; a first-degree familial history of heart attack and/or sudden death at the age of <55 years; surgical, musculoskeletal, osteoarticular, and/or neurological pathology; hydro-electrolytic disturbance; and electrocardiographic (ECG) abnormalities. Additional exclusion criteria were verbal intelligence quotient (IQ) judged as impaired using neuropsychological evaluation (Mill-Hill Test) [21] and psychotic symptoms (assessed using the MINI 7.0.2 for DSM-5).

### 2.3. Variables

Socio-demographic (age), clinical variables (BMI, illness duration), and the EDI-2 score [22] were collected.

The EDI-2 score, translated into its French version [23], provided a rating of behavioural and psychological dimensions relevant to eating disorders with good internal consistency (0.65 to 0.80 in patients with AN and >0.60 in HC [22,24,25].

The Godin Leisure Time Exercise questionnaire (GLTEQ) [26] was used to assess the amount of physical exercise reported by participants over a 7-day span. According to the frequency and intensity of the exercise practised (light, moderate, vigorous), a score of total weekly metabolic equivalents (METs) expended in physical exercise was obtained.

To assess the psychopathological valence of physical exercise, exercise addiction was screened in the two populations with the “Exercise Addiction Inventory” (EAI) [27]. The EAI is a six-item screening tool assessing symptoms of exercise addiction, with an internal consistency of 0.84 [28]. Each item is rated on a 5-point Likert scale, providing a summed score to distinguish among subjects at risk for exercise addiction (scores of 24 or higher), those with some symptoms (scores between 13 and 23), and those at no risk (with scores between 0 and 12).

To evaluate BID, participants were asked to indicate which image was the closest to their own silhouette among a set of 61 computer-generated body-only pictures of women with methodically manipulated bodies in terms of fatness versus thinness [29]. Each picture was associated with a distinct BMI and put in order of increasing size, ranging from 13.19 to 120.29 kg/m^2^. The advantages of this validated tool reside in the use of visual and realistic images, allowing the manipulation of a single body and the minimisation of attentional biases such as facial features that could draw attention away from the body and nudity [29].

The difference calculated between perceived and real BMI was used as a measure of BID in this study.

### 2.4. Procedure

BID was assessed twice in both groups: (1) at baseline and (2) after a standardised physical effort test.

The standardised effort test was performed on a bicycle connected to specialised software (TACX^®^ system), ensuring the same load of effort for each participant. The test encompassed a warm-up phase, where subjects were asked to ride for 1 kilometre at a speed that felt pleasant, followed by an effort phase, where the power of the device automatically rose by 30 watts every 2 min starting from 60 watts. Subjects were instructed to pull the break when the workload became unbearable. In the last phase, participants had to ride for 15 min at 50% of the power of the last 2-min segment completed in the effort phase.

Immediately after completing the standardised effort phase, participants were asked to rate their body image again. The difference between the BMI perceived after the effort and the real BMI was used as a measure of post-effort BID (Figure 1).

To calculate the variation of BID induced by physical exercise, we computed the difference between post-effort BID and baseline BID.

### 2.5. Statistical Analysis

Normality of distribution was assessed using the inspection of skewness and kurtosis. Values of skewness between −2 and +2, and values of kurtosis between −7 and +7, were considered acceptable to prove normal univariate distribution [30,31].

First, we compared baseline BID between AN and HC groups using Student’s test to validate the basal postulate that BID is significantly higher in AN.

Correlation analyses were performed between baseline BID and exercise measures (GLTEQ and EAI scores) separately in patients and HC. All variables are reported with Pearson’s correlation coefficient (r) and their respective statistical significance, set at *p* < 0.05. To rule out the confounding effect of AN severity and psychopathology on the relationship between BID and physical exercise, we performed correlation analyses between BID and BMI, EDI-2 score, and duration of illness.

To ascertain the existence of a specific, exercise-induced aggravation of BID in AN, we compared exercise-induced BID variation between AN and HC using Student’s test.

All statistical analyses were performed using SPSS (IBM Corp. Released 2012. IBM SPSS Statistics for Macintosh, Version 21.0. IBM Corp., Armonk, NY, USA).

## 3. Results

Forty patients with AN and 24 HC were assessed for baseline BID and declarative exercise measures. Amongst them, 31 patients and 24 HC also performed the standardised effort test and were assessed on post-exercise BID.

As expected, patients with AN reported significantly higher BID than HC (*p* < 0.01).

With comparable weekly amounts of physical exercise, patients with AN showed higher exercise addiction than HC (*p* < 0.01) (Table 1).

In patients with AN, a significant correlation was found between BID and EDI-2 scores (r = 0.39; *p* = 0.01) but not between BID and BMI (r = 0.28; *p* = 0.07), age (r = −0.27; *p* = 0.09), and illness duration (r = −0.03; *p* = 0.87).

In AN only, we found a correlation between GLTEQ and baseline BID scores (r = 0.48, *p* < 0.01), whose significance was preserved after controlling for psychopathological severity (EDI-2 score) (r = 0.47, *p* < 0.01).

After the standardised effort, patients with AN had significantly higher increases of BID than HC, with a difference of almost 4 kg/m^2^ compared to 0.8 kg/m^2^ in healthy HC (*p* < 0.01) (Table 2).

## 4. Discussion

This research had the aim of determining the relationship between physical exercise and a core symptom of BID in AN. A significant correlation between the amount of physical exercise practised and BID was found in patients with AN, who also reported a significant increase in BID after a single bout of exercise. Taken together, these results suggest a specific, deteriorating effect of physical exercise on BID in AN.

In line with previous findings [17], physical exercise resulted in having illness-specific effects in patients with AN, given the absence of a relationship between BID and physical exercise in unaffected participants. This research extends the field of illness-specific effects of physical exercise in AN. Previous research found an AN-specific correlation between cognitive rigidity and the quantity of physical exercise practised [32], providing a sound explanation of the persistence of physical exercise even at advanced stages of the disease. A second illness-specific effect, with endophenotypic characteristics [17], is the mood-regulating power of physical exercise, present even in unaffected relatives of patients with AN to a lesser extent. Moreover, patients experience more pleasure from higher efforts [17] and report higher levels of exercise addiction [17,32]. These lines of evidence suggest that patients with AN have a different attitude towards sport compared to healthy subjects [33], which is partially explained by an alteration of the reward system [34,35,36] and a rigidity of cognitive functioning paving the way to compulsive behaviours [37,38]. However, beyond its reinforcing and pleasurable effects [17,38], physical exercise also seems able to aggravate BID in patients with AN [17] as a sort of double-edged sword, as shown by the results of the present study. These conclusions retrace the pattern of the cognitive-behavioural model of compulsive exercise in AN, indicating weight and shape concerns as central factors in maintaining this symptom [14,16,38]. Additionally, this research suggests that the relationship between BID and compulsive exercise may be bidirectional, with physical exercise being a putative factor in maintaining BID. Hence, the aggravation of a vicious circle where the drive to physical exercise observed in patients with AN is also maintained by its effect on BID since body image perception is worsened, rather than relieved, by physical exercise.

Unsurprisingly, psychopathological severity (EDI-2 score) is positively correlated with BID. However, clinical severity, reflected by BMI, seems to play a limited role in BID. This result serves as an indirect confirmation of the pathological nature of BID, unrelated to thinness, unrelievable by any further weight loss, and resulting from an alteration of perceptual [2], interoceptive [39], and cognitive [37,40] abilities in patients only. Thus, it would not be inappropriate to suppose the existence of an intrinsic vulnerability to BID in AN, especially knowing that the main neurocognitive bases of BID (i.e., weak central coherence [41] and cognitive rigidity or impaired set-shifting [42]) have endophenotypic characteristics in AN, which are trait-associated and stable along the evolution of the illness. The persistence of BID even after weight restoration [43] further supports this assumption. Weak central coherence (i.e., the inability to derive overall meaning/view from a collection of details) may explain the tendency of patients to focus on limited and specific parts of their body rather than on the ensemble, preventing them from having a correct estimation of their degree of emaciation.

The physiopathology of exercise-induced BID is still to be determined. The current research identified some cognitive factors that are common to compulsive exercise and BID, such as cognitive rigidity or perfectionism. The impairment of cognitive flexibility could translate into the incapacity to update one’s body image over time despite progressive emaciation, and rigid and persistent perceptual illusions of body image in AN may not change even after weight loss [44]. Unhealthy preoccupations with body shape lead to intensive focus on the body and the search for perfection, which is typical of rigid personalities [45,46,47]. Now, rigidity and perfectionism are two of the main maintaining factors in maladaptive weight loss behaviours, like food restriction [48] and compulsive physical exercise [15,38]. Furthermore, a correlation between cognitive rigidity and physical exercise has already been identified as a specific feature of AN [32]. Hence, cognitive rigidity could represent a sound mediator of the relationship between physical exercise and BID, but further research is needed to confirm this relationship.

Another mechanism linking physical exercise to BID could be dysfunctional interoception [39]. Besides hunger and satiety regulation, and decision-making ability, interoception is involved in emotional awareness and body image perception. Many aspects of compulsive exercise suggest a role for dysfunctional interoception in the development of this maladaptive behaviour. For instance, compulsive exercise is often denied or unrecognised by patients with AN [49], who tend to underestimate the quantity of physical activity they practice. If this denial can be linked to a fear of stigmatisation, poor insight and self-awareness are two important factors accounting for the gap between the objective and the perceived amount of physical exercise of patients. Difficulties with interoception at an emotional level are a recognised component of the drive to physical exercise to regulate mood and emotions in eating-disordered individuals [17,50]. This impairment of interoception could finally explain the alteration of the physical sensation related to exercise. A mismatch between subjective experiences and objective physiological responses has been pinpointed in previous research [50], suggesting abnormal integration between reported and actual interoceptive states. This reduced ability to accurately perceive bodily signals may thus translate into both altered sensations related to body image and an aberrant perception of the stress induced on the organism by physical exercise in energy-deficiency conditions.

To confirm these hypotheses, future research should address the effects of physical exercise on interoception in patients with AN. To our knowledge, a previous study explored the effects of the practice of group Qigong on interoception in patients with AN [51], with beneficial effects in reducing the cleavage between body and mind. However, while this practice is based on the principles of stillness, letting go, openness, and relaxation, the characteristics of physical exercise practised by patients with AN are the opposite. Patients tend to privilege a solitary practice that is repetitive, automatic, and costly in terms of caloric expense [15,38]. As a habit-driven and compulsive behaviour [9,48], this practice instead promotes a dissociation between body and mind, which could favour the inability of patients correctly to perceive their bodies after a physical effort.

Further paths of future research should focus on the metabolic, emotional, and neurological mediation of the effects of physical exercise on BID.

The major strengths of this study are represented by its design, allowing an experimental demonstration of an AN-specific increase in BID after exercise, and by the choice of modern, precise, and validated tools to assess BID and physical exercise. Our results have the double asset of replicating pre-existing research by using more precise assessment tools of BID and reporting the important finding of radically different relationships between body image perception and physical exercise between patients with AN and HC.

The present study presents several limitations, such as a relatively small sample size, a scarce number of HC, and the impossibility of performing some tests on the entire population (for instance, the standardised effort test was not performed by some patients with AN because of the presence of severe and/or acute complications of malnutrition, such as hydro-electrolytic imbalance, electrocardiographic abnormalities, or severe bradycardia). A major limitation of this study is the lack of a test-retest condition not involving a standardised effort test between the two assessments of BID. This would have provided further support to the reliability of this approach to BID assessment and to the determinant, causative role of physical exercise in the aggravation of BID in patients with AN. Nevertheless, the inclusion of a control group, showing limited to absent changes in body image perception after a physical effort, provides an indirect approach to confirm the psychometric validity of the protocol.

Another limitation is represented by the lack of distinction between AN restricting and binge eating/purging subtypes. Although it would have been possible to identify subgroups of participants with AN to search subtype-specific patterns of the relationship between BID and physical exercise, the relatively small number of participants precluded such analyses.

The findings of this research may have an interesting clinical impact. It could be assumed that targeting patients’ attitudes toward physical exercise could be instrumental in helping to rehabilitate their own body image perception. Correct management of unhealthy physical exercise in AN could partially restore, or at least limit, the further aggravation of BID in AN. An important challenge for practitioners is rebalancing the relationship between physical exercise practice and BID. The rehabilitation towards a healthy and beneficial practice of physical exercise is crucial to remission from AN. The choice of the most appropriate exercise programmes in the management of patients with AN should take into account their effects on body image perception and its cognitive and emotional correlates. To achieve this goal, multidisciplinary approaches should be promoted, targeting at once the improvement of BID and of the attitude of patients towards physical exercise. Both dimensions should be regularly monitored using the appropriate tools in order to assess the efficacy of treatment strategies and to prevent relapses.

## 5. Conclusions

The findings of this research aim to further disentangle the mechanisms supporting the development and maintenance of AN, suggesting an illness-related effect of physical exercise on BID in patients with AN. This exercise-induced aggravation of BID could explain the pursuit of this spiral despite ongoing malnutrition. Effective therapeutic strategies should restore a healthy attitude towards physical exercise and monitor the effect of the latter on body image perception and bodily sensations.

## Figures and Tables

**Figure 1 nutrients-14-04507-f001:**
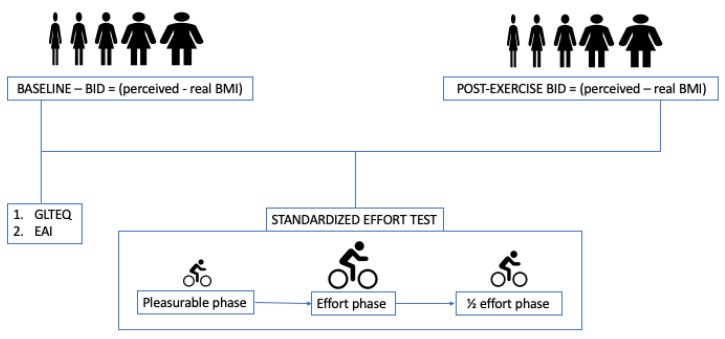
Protocol of the study. Participants were invited to choose the silhouette resembling their own body image before and after a standardised effort test aimed at obtaining the same amount of effort from all participants. BID = body image distortion. BMI = Body Mass Index. GLTEQ = Godin Leisure Time Exercise Questionnaire. EAI = Exercise Addiction Inventory.

**Table 1 nutrients-14-04507-t001:** Characteristics of the population of patients with anorexia nervosa and healthy controls. Standard deviations are reported in brackets.

	AN	Healthy Controls	Student’s t	DF	*p*-Value
Clinical and socio-demographic variables					
Age (years)	23.42 (5.59)	25.50 (3.16)	−1.89	61	0.06
BMI (kg/m^2^)	16.78 (2.31)	21.82 (3.08)	−7.45	62	<0.01
EDI-2 total score	97.29 (46.52)	28.08 (22.71)	8.03	61	<0.01
Illness duration (years)	7.32 (2.34)	-	-	-	-
Physical exercise					
GLTEQ score (MET equivalents)	63.74 (60.21)	57.33 (29.54)	0.49	64	0.63
EAI score	17.56 (7.16)	13.58 (4.55)	2.74	62	0.01

BMI = Body Mass Index. EDI-2 = Eating Disorder Inventory-2. GLTEQ = Godin Leisure Time Activity Questionnaire. EAI = Exercise Addiction Inventory. AN = Anorexia Nervosa. MET = metabolic equivalents of task.

**Table 2 nutrients-14-04507-t002:** Body image distortion at baseline and after a physical effort in patients with anorexia nervosa and healthy controls. Standard deviations are reported in brackets.

	AN	Healthy Controls	Student’s t	DF	*p*-Value
Body image distortion (BID)					
Self-rated BMI before exercise	19.016 (4.73)	21.957 (3.91)	2.57	63	<0.01
Baseline BID (BMI points: kg/m^2^)	3.031 (4.84)	0.136 (1.91)	3.43	58	<0.01
Self-rated BMI after physical exercise	22.985 (5.95)	22.749 (4.83)	1.21	52	0.23
Physical exercise induced-BID (BMI points)	3.969 (4.46)	0.792 (2.86)	3.24	52	<0.01

BID = body image distortion.

## Data Availability

The data that support the findings of this study are available from the corresponding author upon reasonable request.

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
