# Peer review of "The Fitter I Am, the Larger I Feel—The Vicious Circle of Physical Exercise in Anorexia Nervosa"

_nutrients, 2022, doi:10.3390/nu14214507_

Round 1
Reviewer 1 Report
It is an interesting study about the relationship between body image distortion and physical exercise in AN.
My main concern is the lack of test and retest reliability assessment in the studied population, outside the experimental conditions (i.e. without physical effort). Without this, we cannot conclude unequivocally that it was physical exercise that changed the perception of female's own body.
Author Response
We sincerely thank the referee for this important comment.
We agree with this criticism, i.e. the lack of a test-retest condition enabling us to confirm the effect of physical exercise as a causative factor for body image distortion aggravation. We identify this lack as a limitation of the study, we thus modified the manuscript in the discussion section (lines 315-321), by introducing the following paragraph :
"A major limitation of this study is the lack of a test-retest condition involving no standardized effort test between the two assessments of BID. This would have provided further support to the reliability of this approach of BID assessment, and to the determinant, causative role of physical exercise in the aggravation of BID in patients with AN. Nevertheless, the inclusion of a control group, showing limited to absent changes of body image perception after a physical effort, provides an indirect approach to confirm the psychometric validity of the protocol."
Reviewer 2 Report
AN has been a subject that researchers have been working on for a long time, which does not make it less important. I thank the researchers for choosing this theme and for being interested in approaching the disease in order to find solutions that help adolescents to escape from these problems. The work is well thought out methodologically, it is easy to read and can help researchers to initiate work in this line that, together with physical activity, manage to prevent these disorders in young people.
Author Response
We really thank the referee for his/her positive comment and encouragements. In line with the referee's suggestions, we will check the references we cited and provide updates where necessary.